# DenseGrounding: Improving Dense Language-Vision Semantics for Ego-centric 3D Visual Grounding

**Henry Zheng**[1][*], **Hao Shi**[1][*], **Qihang Peng**[1], **Yong Xien Chng**[1], **Rui Huang**[1],
**Yepeng Weng**[2], **Zhongchao Shi**[2], **Gao Huang**[1][✉]
[1]Department of Automation, BNRist, Tsinghua University  [2]AI Lab, Lenovo Research
{jh-zheng22,shi-h23,pqh22,chngyx10,hr20}@mails.tsinghua.edu.cn
{wengyp1,shizc2}@lenovo.com
{gaohuang}@tsinghua.edu.cn

## Abstract

Enabling intelligent agents to comprehend and interact with 3D environments through natural language is crucial for advancing robotics and human-computer interaction. A fundamental task in this field is ego-centric 3D visual grounding, where agents locate target objects in real-world 3D spaces based on verbal descriptions. However, this task faces two significant challenges: (1) loss of fine-grained visual semantics due to sparse fusion of point clouds with ego-centric multi-view images, (2) limited textual semantic context due to arbitrary language descriptions. We propose DenseGrounding, a novel approach designed to address these issues by enhancing both visual and textual semantics. For visual features, we introduce the Hierarchical Scene Semantic Enhancer, which retains dense semantics by capturing fine-grained global scene features and facilitating cross-modal alignment. For text descriptions, we propose a Language Semantic Enhancer that leverages large language models to provide rich context and diverse language descriptions with additional context during model training. Extensive experiments show that DenseGrounding significantly outperforms existing methods in overall accuracy, with improvements of 5.81% and 7.56% when trained on the comprehensive full dataset and smaller mini subset, respectively, further advancing the SOTA in ego-centric 3D visual grounding. Our method also achieves **1st place** and receives **Innovation Award** in the CVPR 2024 Autonomous Grand Challenge Multi-view 3D Visual Grounding Track, validating its effectiveness and robustness. Code

## 1 Introduction

Recent years have seen increasing attention in the field of embodied AI, with the introduction of 3D visual grounding benchmarks (Achlioptas et al., 2020a; Chen et al., 2020; Wang et al., 2024) prompting a surge of research works (Guo et al., 2023; Wu et al., 2023; Zhao et al., 2021; Jain et al., 2022; Yang et al., 2024; Huang et al., 2022; Wang et al., 2024). The task of 3D visual grounding, which aims to locate target objects in real-world 3D environments based on natural language descriptions, is a fundamental perception task for embodied agents. This capability is crucial for enabling agents to interact with and understand their surroundings through language, facilitating applications in robotics and human-computer interaction. For instance, accurate 3D visual grounding empowers robots to perform tasks such as object retrieval and manipulation based on verbal instructions, enhancing their functionality in service and assistive scenarios.

Despite these advances, significant challenges continue to hinder the performance of 3D perception systems. One major challenge lies in how embodied agents perceive their environment, as they typically rely on ego-centric observations from multiple views while moving around, lacking a holistic, scene-level perception. Some methods attempt to enhance scene-level understanding using reconstructed scene-level 3D point clouds (Wu et al., 2023; Zhao et al., 2021; Jain et al., 2022; Yang

---

[*]Equal contributions. [✉]Corresponding author.

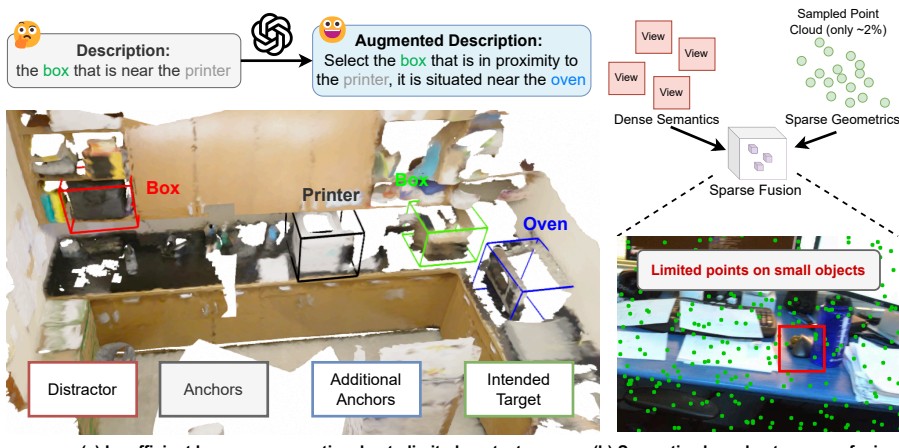

(a) Insufficient language semantics due to limited context. (b) Semantics loss due to sparse fusion.

Figure 1: (a) illustrates how limited context due to arbitrary descriptions leads to insufficient language semantics. (b) highlights the issue of losing fine-grained semantics in sparse fusion.

et al., 2024; Guo et al., 2023; Huang et al., 2022; 2024b;c) following the previous 3D perception methods (Wu et al., 2024b; Jiang et al., 2020), but these approach are impractical in real-world applications where such comprehensive scene-level information is not readily available. In this context, EmbodiedScan (Wang et al., 2024) have emerged, utilizing multi-view ego-centric RGB-D scans to address the need for models to understand scenes directly from sparse views. It decouples the encoding of RGB images and depth-reconstructed point clouds from ego-centric views to extract both semantic and geometric information. Semantic features are then projected onto 3D space using intrinsic and extrinsic matrices to create a semantically enriched sparse point cloud for bounding box regression. However, due to the high number of points in the reconstructed point cloud and computational limitations, only a sparse subset (around 2%) is sampled. This sampling results in a significant loss of visual semantics, limiting the model's ability to retain fine-grained object details, especially for smaller objects, and ultimately hindering grounding performance.

Another challenge is the ambiguity in natural language descriptions found in existing datasets (Chen et al., 2020; Achlioptas et al., 2020a; Wang et al., 2024). Humans naturally provide concise instructions, often lacking critical reference points or anchor objects, which leads to inherent ambiguities. For example, a command like "Please get the water bottle on top of the table" does not specify which table or provide additional context in environments with multiple similar objects. This conciseness makes it difficult for models to disambiguate between similar objects within complex scenes (Wang et al., 2024). Furthermore, the increased complexity of scenes in new datasets—such as a more similar objects instances in a single scene—exacerbates this issue. In result, training on these data may cause model to have suboptimal performance. While previous attempts have been made to enrich descriptions using large language models with predefined templates (Guo et al., 2023) or by concatenating related descriptions (Wang et al., 2024), these methods have not fully mitigated the ambiguities present in the annotations and have not utilize the available informations in the dataset.

Addressing these challenges is crucial for advancing the field of embodied AI perception and enhancing the practical utility of 3D visual grounding in real-world applications. In response to these challenges, we propose DenseGrounding, a novel method for multi-view 3D visual grounding that alleviates the sparsity in both visual and textual features. Specifically, to address the loss of fine-grained visual semantics, we introduce the *Hierarchical Scene Semantic Enhancer* (HSSE), which enriches visual representations with global scene-level semantics. HSSE operates hierarchically, starting with *view-level semantic aggregation* that integrates multi-scale features from RGB images within each view to capture detailed view-level semantics. Then, HSSE performs *scene-level semantic interaction* to combine these aggregated multi-view semantics with language features from descriptions, promoting cross-modal fusion and a hoslistic understanding of the scene. Finally, HSSE conducts *semantic broadcast* that infuses the enriched global scene-level semantics into the multi-scale feature maps of depth-reconstructed point cloud, improving their fine-grained semantics.

To mitigate the ambiguity in natural language descriptions used for training, for the text modality, we propose the *Language Semantic Enhancer* (LSE) that constructs a scene information database based on the existing dataset, EmbodiedScan. This database captures crucial information about object relationships and locations within the scene. During training, the LSE leverages LLM to enriches input descriptions with additional anchors and context, reducing confusion among similar objects and providing more robust representations by incorporating external knowledge. This enrichment increases the semantic richness of the language descriptions and reduces inherent ambiguities.

By addressing the ambiguities in annotated text and enhancing the model's capacity to capture scene-level global visual semantics from ego-centric multi-view inputs, DenseGrounding, advances the state-of-the-art in 3D visual grounding. Extensive experiments demonstrate the effectiveness of our approach, highlighting the importance of enriched textual descriptions and bidirectional interaction between text and visual modalities in overcoming the limitations of previous methods.

Our main contributions are as follows:

- We develop a novel Language Semantic Enhancer (LSE) that employs LLM-assisted description augmentation to increase semantic richness and significantly reduce ambiguities in natural language descriptions. By leveraging an LLM grounded in a scene information database, our approach enriches the diversity and contextual clarity of the textual features.

- We introduce a Hierarchical Scene Semantic Enhancer (HSSE) that effectively aggregates ego-centric multi-view features and captures fine-grained scene-level global semantics. HSSE enables interaction between textual and visual features, ensuring the model captures both global context and detailed object semantics from ego-centric inputs.

- Our method achieves state-of-the-art performance on the EmbodiedScan benchmark, substantially outperforming existing approaches. We secured first place in the CVPR 2024 Autonomous Driving Grand Challenge Track on Multi-View 3D Visual Grounding (Zheng et al., 2024), demonstrating the practical effectiveness and impact of our contributions.

## 2 RELATED WORKS

**3D Visual Grounding.** The integration of language and 3D perception for scene understanding is crucial in embodied AI, enabling agents to navigate, understand, and interact with complex real-world environments. Key tasks (Chen et al., 2020; Achlioptas et al., 2020b; Azuma et al., 2022; Chen et al., 2021) in this domain, 3D Visual Grounding (3DVG) (Wu et al., 2023; Jain et al., 2022; Zhu et al., 2023b; Huang et al., 2022; 2024a; Cai et al., 2022), 3D Captioning (Huang et al., 2024a; Chen et al., 2023; Cai et al., 2022) and 3D Question Answering (3DQA) (Huang et al., 2024a), combine natural language understanding with spatial reasoning to advance multimodal intelligence in 3D spaces. Our work focuses on ego-centric 3D visual grounding tasks that integrate multimodal data to localize objects in 3D space.

Major 3D visual grounding methods can be divided into one-stage and two-stage architectures. One-stage approaches (Chen et al., 2018; Liao et al., 2020; Luo et al., 2022; Geng & Yin, 2024; He & Ding, 2024) fuse textual and visual features in a single step, enabling end-to-end optimization and faster inference. Recent works have optimized single stage methods in various ways: 3D-SPS (Luo et al., 2022) progressively selects keypoints guided by language; ViewInfer3D (Geng & Yin, 2024) uses large language models (LLMs) to infer embodied viewpoints; and RefMask3D (He & Ding, 2024) integrates language with geometrically coherent sub-clouds via cross-modal group-word attention, producing semantic primitives that enhance vision-language understanding.

Two-stage approaches (Yang et al., 2019; Achlioptas et al., 2020a; Huang et al., 2022; Guo et al., 2023; Wu et al., 2024a; Chang et al., 2024) first use pre-trained object detectors (Jiang et al., 2020; Wu et al., 2024b) to generate object proposals or utilize ground truth bounding boxes, which are then matched with the linguistic input. Advances in this framework include the Multi-View Transformer (Huang et al., 2022), projecting the 3D scene into a holistic multi-view space; ViewRefer (Guo et al., 2023), leveraging LLMs to expand grounding texts and enhancing cross-view interactions with inter-view attention; and MiKASA Transformer (Chang et al., 2024), introducing a scene-aware object encoder and a multi-key-anchor technique to improve object recognition and spatial understanding. In our paper, we target the more challenging setting of single-stage methods.

**LLMs for Data Augmentation.** Recent advances in large language models (LLMs) (Achiam et al., 2023; Touvron et al., 2023; Du et al., 2021) have demonstrated remarkable capabilities. Fully leveraging the power of LLMs beyond language processing, some recent studies have successfully integrated them with other modalities, leading to the development of highly effective multi-modal methods (Moon et al., 2022; Guo et al., 2023; Feng et al., 2024) for vision-language tasks.

However, directly utilizing LLM or VLMs in robotics perception tasks remains a research question (Kim et al., 2024; Zhen et al., 2024; Phan et al., 2024). Therefore, leveraging LLMs to directly implement *data augmentation* and maximizing their potential to enhance text data diversity become a new trend (Peng et al., 2023; Khan et al., 2023; Dunlap et al., 2023; Guo et al., 2023; Ding et al., 2024). Among them, ALIA (Dunlap et al., 2023) leverages LLMs to generate image descriptions and guide language-based image modifications, augmenting the training data. This approach surpasses traditional data augmentation techniques in fine-grained classification, underscoring the value of LLM-driven augmentation in visual tasks. In 3D visual grounding, although ViewRefer (Guo et al., 2023) employs LLMs to augment textual data through rephrasing view-dependent descriptions from varying perspectives, however the data samples are augmented individually and suffer from limited textual semantic information. To obtain high-quality description augmentations that disambiguate among similar items and more robust representations, we utilize LLMs to enrich input descriptions utilizing readily available information in the dataset as additional context.

## 3 PRELIMINARIES

### 3.1 EGO-CENTRIC 3D VISUAL GROUNDING

In real-world scenarios, intelligent agents perceive their environment without any prior scene knowledge. They often rely on ego-centric observations, such as multi-view RGB-D images, rather than pre-established scene-level priors like pre-reconstructed 3D point clouds of the entire scene, as commonly used in previous studies (Wu et al., 2024b; Huang et al., 2024b; Chng et al., 2024).

Following Wang et al. (2024), we formalize the ego-centric 3D visual grounding task as follows: Given a language description $L \in \mathbb{R}^T$, together with $V$ views of RGB-D images $\{(I_v, D_v)\}_{v=1}^V$, where $I_v \in \mathbb{R}^{H \times W \times 3}$ represents the RGB image and $D_v \in \mathbb{R}^{H \times W}$ denotes the depth image of the $v$-th view along with their corresponding sensor intrinsics $\{(K_v^I, K_v^D)\}_{v=1}^V$ and extrinsics $\{(T_v^I, T_v^D)\}_{v=1}^V$, the objective is to output a 9-degree-of-freedom (9DoF) bounding box $B = (x, y, z, l, w, h, \theta, \phi, \psi)$. Here, $(x, y, z)$ are the 3D coordinates of the object's center, $(l, w, h)$ are its dimensions, and $(\theta, \phi, \psi)$ are its orientation angles. The task is to determine $B$ such that it accurately corresponds to the object described by $L$ within the scene represented by $\{(I_v, D_v)\}_{v=1}^V$.

### 3.2 OVERVIEW OF THE ARCHITECTURE

We adopt and improve upon the previous SOTA in ego-centric 3D visual grounding (Wang et al., 2024). To prevent geometric information from interfering with semantic extraction and fully leverage each modality's strengths, Wang et al. (2024) decouple the encoding of input RGB and depth signals from each ego-centric views, following Liu et al. (2023). Specifically, the depth data from each view is transformed into a partial point cloud, which is then integrated into a holistic 3D point cloud $P \in \mathbb{R}^{N \times 3}$ using global alignment matrices, thereby preserving precise geometric details.

Next, a semantic encoder and a geometric encoder extract multi-scale semantic and geometric features from $\{I_v\}_{v=1}^V$ and $P$, respectively, denoted as $\{F_{\text{Sem}}^s\}_{s=1}^S \in \mathbb{R}^{V \times H_s \times W_s \times C_s}$ and $\{F_{\text{Geo}}^s\}_{s=1}^S \in \mathbb{R}^{N \times C_s}$. Here, $N$ is the total number of points sampled from all perspective views, and $S$ is the number of scale for the encoded feature maps.

The semantic features from RGB data are then lifted to the 3D space using intrinsic and extrinsic matrices $\{(K_v^I, T_v^I)\}_{v=1}^V$, and concatenated with geometric features at multiple scales, followed by fusion through a Feature Pyramid Network (FPN). However, due to the large number of points in the depth-reconstructed point cloud and computational limitations, only a sparse subset (about 2%) is sampled, and the semantic features at unsampled locations are discarded, resulting in significant semantic information loss. To address this issue, we introduce a Hierarchical Scene Semantic Enhancer module (detailed in Sec. 4.1) to improve scene-level semantics. The enhanced features are

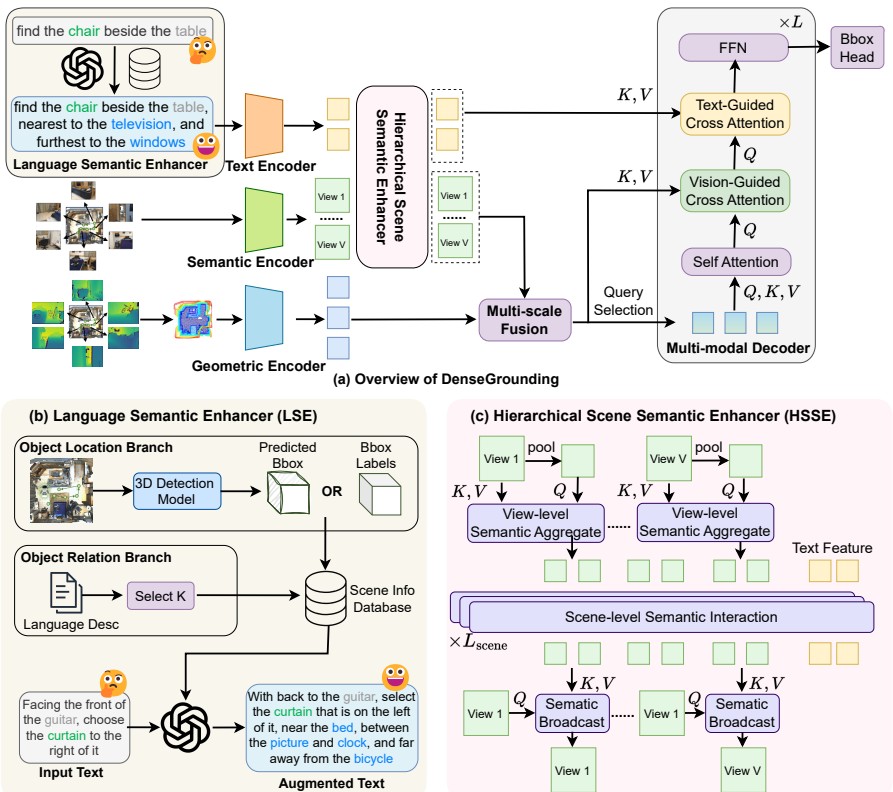

Figure 2: (a) shows overall framework, while (b) details the Language Semantic Enhancer (LSE) module, and (c) describes the Hierarchical Scene Semantic Enhancer (HSSE) module.

then fed into the decoder, which remains unchanged from the previous work, where the visual tokens most similar to the text features are selected as queries.

# 4 METHOD

In this section, we present our proposed method, DenseGrounding, for ego-centric 3D visual grounding. As shown in Figure 2, our method consists of three key components: Hierarchical Scene Semantic Enhancer (Sec. 4.1), LLM-based Language Semantic Enhancer (Sec. 4.2), and an Enhanced Baseline model (Sec. 4.3). We will describe each component in detail in the following subsections.

## 4.1 HIERARCHICAL SCENE SEMANTIC ENHANCER

To mitigate the semantic loss inherent in sparse point clouds, we introduce the Hierarchical Scene Semantic Enhancer (HSSE) module, which extracts individual view semantics and enriches visual representation with global scene-level semantics. This enriched information is then unprojected to the depth reconstructed point cloud during fusion, minimizing the semantic loss.

Our proposed HSSE module works in a hierarchical manner, starting with view-level semantic aggregation to capture view-level semantics from individual egocentric perspective views. Then, it fuses these aggregated view semantics with language semantics, facilitating scene-level multi-view semantic interaction and cross-modal feature fusion. Moreover, a broadcast module integrates these scene-level semantics into multi-scale semantic feature maps, enriching the fine-grained semantics in the sparse point cloud. Finally, the HSSE employs an aggregation-broadcast mechanism to filter out irrelevant information, ensuring efficient and streamlined semantic transfer to the point cloud for enhanced performance.

**View-Level Semantic Aggregation.** With the encoded 2D features from each view, HSSE performs view-level semantic aggregation to capture view-level global semantics within each view. For the $v$-th view, given the multi-scale semantic features $\{F_{\text{Sem}}^{v,s}\}_{s=1}^S \in \mathbb{R}^{H_s \times W_s \times C_s}$ extracted from RGB image, we aim to capture the salient semantic cues in each view.

First, these semantic features are processed through a Feature Pyramid Network (FPN) to integrate information across different scales. A specific scale $s$ of the feature map is then selected as the reference feature $F_{\text{Ref}}^v \in \mathbb{R}^{H_s \times W_s \times C_s}$.

$$F_{\text{Ref}}^v = \text{Conv}\left(\sum_{i=1}^{s} \text{Upsample}(F_{\text{Sem}}^{v,i})\right) \in \mathbb{R}^{H_s \times W_s \times C_s} \tag{1}$$

Next, adaptive average pooling is applied to the reference feature to reduce its spatial dimensions, yielding the initial query features $F_Q^v \in \mathbb{R}^{h_s \times w_s \times C_s}$.

$$F_Q^v = \text{Pool}(F_{\text{Ref}}^v) \in \mathbb{R}^{h_s \times w_s \times C_s} \tag{2}$$

Subsequently, we compute the cross-attention between the pooled queries and the reference feature to aggregate semantic information at the view level.

$$\hat{F}_Q^v = \text{CrossAttn}(Q = F_Q^v, K = F_{\text{Ref}}^v, V = F_{\text{Ref}}^v) \tag{3}$$

where $\hat{F}_Q^v \in \mathbb{R}^{h_s \times w_s \times C_s}$ encapsulates the global semantics of the view. We then apply self attention layer, to further refine the features and model intra-view relationships.

$$\hat{F}_Q^v = \text{SelfAttn}(Q = \hat{F}_Q^v, K = \hat{F}_Q^v, V = \hat{F}_Q^v) \tag{4}$$

**Scene-Level Semantic Interaction.** Moreover, HSSE conducts scene-level semantic interaction by integrating the aggregated multi-view semantics $\{\hat{F}_Q^v\}_{v=1}^V$ with the language semantics $F_{\text{Lang}}$ derived from the language description. To promote cross-modal fusion and scene-level interaction among multi-view semantics, we concatenate $\{\hat{F}_Q^v\}_{v=1}^V$ with $F_{\text{Lang}}$ and apply $L_{\text{scene}}$ layers of self-attention.

$$[\{\hat{F}_Q^{v'}\}_{v=1}^V, F_{\text{Lang}}'] = \text{SelfAttn}^{L_{\text{scene}}}([\{\hat{F}_Q^v\}_{v=1}^V; F_{\text{Lang}}]), \tag{5}$$

where $\{\hat{F}_Q^{v'}\}_{v=1}^V$ captures the scene-level global semantics, and $F_{\text{Lang}}'$ represents the refined language semantics after the interaction.

**Semantic Broadcast.** Finally, HSSE broadcasts the scene-level global semantics $\hat{F}_Q^{v'}$ back to the multi-scale semantic feature maps $\{F_{\text{Sem}}^{s,v}\}_{s=1}^S \in \mathbb{R}^{H_s \times W_s \times C_s}$ for each view. Specifically, we compute cross-attention between the spatial features $\{F_{\text{Sem}}^{s,v}\}_{s=1}^S$ and the scene-level semantics $\hat{F}_Q^{v'}$, treating the spatial features as queries.

$$\{\hat{F}_{\text{Sem}}^{s,v}\}_{s=1}^S = \text{CrossAttn}(Q = \{F_{\text{Sem}}^{s,v}\}_{s=1}^S, K = \hat{F}_Q^{v'}, V = \hat{F}_Q^{v'}) \tag{6}$$

where $\{\hat{F}_{\text{Sem}}^{s,v}\}_{s=1}^S$ represents the enhanced feature maps enriched with scene-level global semantics.

## 4.2 LLM-based Language Semantic Enhancer

Human interactions with intelligent agents often involve casual and vague language descriptions, resulting in input that lacks clear anchors, limited textual context, and ambiguities. To address this, we propose a Language Semantic Enhancement (LSE) pipeline based on Large Language Models (LLMs) to enhance the training data. This pipeline enriches input language descriptions by leveraging LLMs, which are grounded with a Scene Information Database (SIDB) containing object locations and relationships, providing more contextual details of the corresponding scene.

**Construct Scene Information Database (SIDB).** Visual grounding datasets are typically built on indoor 3D detection datasets, where collected RGB-D images or point clouds are used to represent a scene. Each object within the scene is annotated with 3D bounding boxes, which visual grounding datasets extend by adding language descriptions detailing the object's relationship to its surroundings. To ensure accurate contextualization for LLMs augmenting these language descriptions, we

propose to construct a Scene Information Database (SIDB) based on the annotated training set that is also used by all baseline methods for model training.

For each 3D scene, we follow the design of the datasets and assume that every language description targets a specific bounding box. These descriptions are grouped according to their corresponding scenes, allowing the SIDB to store detailed object relationships within each scene. This approach enables LLMs to better understand the spatial and relational dynamics of objects in a given scene, leading to a more holistic comprehension of the environment.

Moreover, to provide location information about the object instances in the scene, we enrich each of the descriptions in SIDB with position information from the original dataset annotations or pseudo-labels from the detection model, associating each described object with its spatial location. These descriptions, combining both spatial and semantic information, form the backbone of the SIDB. By providing this structured and detailed knowledge, the SIDB allows LLMs to generate more contextually informed and reliable text descriptions grounded in the specific dynamics of each scene.

**Prompt for LLM-based Enhancement.** To effectively enrich the input descriptions, we design a prompt for the LLM that leverages the SIDB to generate more detailed and contextually rich descriptions. By instructing the LLM to utilize positional relationships among objects in the scene, the prompt ensures sufficient and reliable anchors, encouraging the description of the target object using multiple reference objects to minimize ambiguity and distractions. For instance, it draws on spatial context from the database to more accurately reflect the object's relationships to surrounding items. Moreover, the prompt guides the LLM to maintain consistency between the original and augmented text by preserving the original content while adding additional information, which reduces the risk of the generated text deviating from the initial prompt.

**Enhance Description with LLM and SIDB.** For each description to be augmented (raw description), we select the $k$ most related context descriptions in the corresponding scene based on object names from the SIDB and provide them to the LLM. Details on the context description selection can be found in Appendix A.5.2. The LLM leverages the database to incorporate spatial and semantic details, aiming to enrich the text with additional contextual anchors that can help prevent confusion with similar-looking objects.

### 4.3 ENHANCED BASELINE

In this work, we introduce a strong baseline by building upon the state-of-the-art method established by EmbodiedScan for ego-centric multiview 3D visual grounding and further boosting its performance. More details about the proposed improvements can be found in the Appendix A.1.1.

## 5 EXPERIMENTS

**Dataset and Benchmark.** The EmbodiedScan dataset (Wang et al., 2024), used in our experiments, is a large-scale, multi-modal, *ego-centric* dataset for comprehensive 3D scene understanding. It consists of 5,185 scene scans sourced from well-known datasets such as ScanNet (Dai et al., 2017), 3RScan (Wald et al., 2019), and Matterport3D (Chang et al., 2017). This extensive dataset provides a diverse and rich foundation for 3D visual grounding tasks, offering broader scene coverage compared to prior datasets. This makes the dataset significantly larger and more challenging than previous ones, providing a more rigorous benchmark for 3D visual grounding tasks.

For benchmarking, the official dataset maintains a non-public test set for the test leaderboard and divides the original training set into new subsets for training and validation. In this paper, we refer to these as the training and validation sets, while the non-public test set is called the testing set.

**Experimental Settings.** Due to resource limitations, we reserve the full training dataset for baseline comparisons on the test set and leaderboard submissions to ensure a fair and comprehensive evaluation. For the "mini" data in the Data column of Table 1 and analysis experiments in Sec. 5.2, we use a smaller subset of the data as a proxy task in performing experiments. The subset is referred to as mini sets, available through the official release by Wang et al. (2024).

Table 1: Validation Result: Accuracy performance of the models on the official full validation set. We follow the experimental setting proposed by Wang et al. (2024) and use the RGB-D as visual input. The 'Data' column indicates the data split used. [†] denotes the improved baseline discussed in Sec. 4.3. The results reported for EmbodiedScan are obtained by re-evaluating the validation set using the officially provided weights.

| Method | Data | Easy $ACC_{25}$ | Hard $ACC_{25}$ | Indep $ACC_{25}$ | Dep $ACC_{25}$ | Overall $ACC_{25}$ |
|---|---|---|---|---|---|---|
| ScanRefer (Chen et al., 2020) | Full | 13.78 | 9.12 | 13.44 | 10.77 | 12.85 |
| BUTD-DETR (Jain et al., 2022) | Full | 23.12 | 18.23 | 22.47 | 20.98 | 22.14 |
| L3Det (Zhu et al., 2023a) | Full | 24.01 | 18.34 | 23.59 | 21.22 | 23.07 |
| EmbodiedScan (Wang et al., 2024) | Full | 39.28 | 30.88 | 38.23 | 39.30 | 38.60 |
| **DenseGrounding** | Full | **45.31** | **34.23** | **44.40** | **44.42** | **44.41** |
| | | (+6.03) | (+3.35) | (+6.17) | (+5.12) | (+5.81) |
| EmbodiedScan (Wang et al., 2024) | Mini | 34.09 | 30.25 | 33.56 | 34.20 | 33.78 |
| EmbodiedScan[†] | Mini | 36.23 | 30.51 | 35.89 | 34.30 | 35.77 |
| | | (+2.14) | (+0.26) | (+2.33) | (+0.10) | (+1.99) |
| **DenseGrounding** | Mini | **41.95** | **34.38** | **40.89** | **42.19** | **41.34** |
| | | (+7.86) | (+4.13) | (+7.33) | (+7.99) | (+7.56) |

We report accuracy using the official metric, considering instances where IoU exceeds 0.25. Additionally, we present results for "Easy" and "Hard" scenes, classifying a scene as "Hard" if it contains 3 or more objects of the same class. The "Dep" and "Indep" metrics further challenge spatial understanding ability by assessing its performance with and without perspective-specific descriptions.

**Implementation Details.** We follow EmbodiedScan (Wang et al., 2024) by using feature encoders, sparse fusion modules, and a DETR-based (Carion et al., 2020) decoder. Specifically, we use ResNet50 (He et al., 2016) and MinkNet34 (Choy et al., 2019) as the 2D and 3D vision encoders, respectively. Moreover, as mentioned in Sec. 4.3, we replace RoBERTA with CLIP (Radford et al., 2021) text encoder as language feature encoders. We first pre-train our image and point cloud encoders on the 3D object detection task, adhering to EmbodiedScan's training settings and integrating CBGS (Zhu et al., 2019). The LSE-augmented text is generated using *GPT4-o mini* and is utilized exclusively during training. For inference, our model processes descriptions directly, without any enhancement, aligning with our baseline methods for fair comparison. More details in App. A.1.2.

## 5.1 MAIN RESULTS

**Evaluation.** We evaluate the 3D visual grounding performance of our proposed method, DenseGrounding, and report the results in Table 1, where we compare it against established SOTA methods from the dataset benchmark. The table is divided into two sections: the upper half presents models trained on the full training set, while the lower half showcases performance using a mini training set, which is approximately 20% of the full dataset.

On the full training set, DenseGrounding achieves a significant improvement of 5.81% over the previous strongest baseline, EmbodiedScan. This substantial gain highlights the effectiveness of our approach in leveraging extensive training data to capture complex visual-linguistic correlations in 3D environments. It demonstrates our method's superior ability to understand and ground linguistic expressions in rich visual scenes. When trained on the more manageable mini-training set, we introduce an enhanced baseline, denoted as EmbodiedScan[†], which achieves a 1.99% performance gain over the original EmbodiedScan. This stronger baseline ensures a more rigorous and fair comparison with our method. Remarkably, even against this enhanced baseline, DenseGrounding attains a substantial 5.57% improvement in overall accuracy, culminating in a total performance gain of 7.56% over the previous state-of-the-art. Specifically, on hard samples, where scenes contain more than three objects of the same class, we observe a 4.14% increase in accuracy. This demonstrates our method's proficiency in accurately identifying the correct target among similar-looking objects.

Table 2: Ablation on LSE. R and L refers to object relationship and object location information in SIDB, respectively.

| Method | Easy | Hard | Overall |
|---|---|---|---|
| Concat Samples | 36.80 | 31.44 | 36.37 |
| LLM | 37.69 | 31.97 | 37.23 |
| LLM+DB(R) | 38.48 | 33.54 | 38.08 |
| LLM+DB(R+L) | 39.29 | 33.55 | **38.82** |

Table 3: Ablation of Proposed Methods. The reported values are Accuracies for predictions greater than 25% IoU with groundtruth.

| LSE | HSSE | Easy | Hard | Overall |
|---|---|---|---|---|
| | | 35.77 | 30.51 | 35.77 |
| | ✓ | 39.58 | 32.60 | 39.02 |
| ✓ | | 39.67 | 34.49 | 39.25 |
| ✓ | ✓ | 41.95 | 34.38 | **41.34** |

As detailed in Table 1, DenseGrounding consistently outperforms the baselines across various evaluation categories, including both easy and hard samples, as well as in view-independent and view-dependent tasks. These consistent gains across different metrics underscore the robustness and generalizability of our approach in 3D visual grounding tasks. Moreover, our model not only achieves the highest performance among all methods but also secured 1st place in the CVPR 2024 Autonomous Driving Grand Challenge Track on Multi-View 3D Visual Grounding.

## 5.2 ANALYSIS EXPERIMENTS

**Effectiveness of Each Component.** We conduct an ablation analysis to assess the effectiveness of each component, as shown in Tab. 3. Our baseline is an enhanced model that excludes text augmentations and semantic enhancements. Introducing the augmented data led to a remarkable accuracy improvement of 3.48%, highlighting its significant impact. The further integration of the HSSE module provided an additional performance boost of 2.09%. Furthermore, the combined use of LSE and HSSE resulted in a 5.57% accuracy improvement on hard samples compared to the baseline, underscoring our model's enhanced capacity for both visual and language understanding, particularly in disambiguation of challenging cases.

**Ablation on LSE.** To evaluate the effectiveness of our proposed LSE, we conducted experiments comparing several text augmentation methods. The "Concat Samples" method from Embodied-Scan disambiguate descriptions by concatenating multiple annotations. "LLM" refers to our re-implementation of template-based LLM augmentation used by Viewrefer (Guo et al., 2023). Our method, "LLM+DB(R)," employs LLM with SIDB incorporating object relationships only, while "LLM+DB(R+L)" extends this approach by adding location information from 3D bounding boxes. It is worth noting that since "Concat Samples" method only has about 25% data, we limit other methods to use 25% data for fair comparisons. The results demonstrate that "LLM+DB(R+L)" achieves the notable over all improvement of 2.45% against naive baseline, confirming the effectiveness of incorporating both object relationships and location data in augmentation process.

**Ablation on HSSE Components.** We further explore the design of the HSSE module through comprehensive experiments, as presented in Tab. 4 and 5. In Tab. 4, we conduct an ablation study to determine the optimal number of self-attention layers needed for effective learning of the scene-feature representation. Additionally, Tab. 5 examines the impact of varying feature map sizes for the pooled $F_Q^v$ feature. The results indicate that a small feature map size may be insufficient to capture the necessary information, while an excessively large feature map may include redundant details. Based on these findings, we adopted the best configuration for our final model.

## 5.3 QUALITATIVE ANALYSIS

We present qualitative results to illustrate the effectiveness of DenseGrounding in improving ego-centric 3D visual grounding performance. Figure 3 shows a comparison between our model and our baseline model, EmbodiedScan. It can be clearly seen that our method outperforms the baseline in correctly identifying the target objects based on ambiguous descriptions. In cases where the baseline model struggles to disambiguate between multiple similar objects, DenseGrounding successfully detects the correct target by leveraging its enriched textual descriptions and robust cross-modal interactions. For instance, in environments where descriptions such as "select the keyboard that is close to

Table 4: Ablation on the number of self attention layers for HSSE.

| $L_{scene}$ | Easy | Hard | Overall |
|---|---|---|---|
| 1 | 40.15 | 35.44 | 39.77 |
| 2 | 40.76 | 32.70 | 40.11 |
| 3 | 41.95 | 34.38 | **41.34** |
| 4 | 41.06 | 33.96 | 40.49 |
| 6 | 40.66 | 33.23 | 40.06 |

Table 5: Ablation on the view feature map size after pooling for HSSE.

| Pooled Size | Easy | Hard | Overall |
|---|---|---|---|
| 1 | 40.61 | 34.91 | 40.15 |
| 3 | 41.92 | 33.75 | 41.26 |
| 5 | 41.95 | 34.38 | **41.34** |
| 7 | 41.92 | 34.28 | 41.30 |
| 9 | 40.05 | 34.60 | 39.61 |

Figure 3: Qualitative Analysis. Comparison of Ground Truth, our baseline and DenseGrounding. Ground truth boxes are shown in green, baseline in red, and DenseGrounding's predictions in blue.

the fan" are provided, our model accurately localizes the keyboard, while the baseline misidentifies nearby objects. This improvement is attributed to HSSE, which captures both fine-grained details and provides better alignment with text, enabling better object identification. These qualitative results demonstrate the enhanced performance of DenseGrounding, especially in complex scenes with multiple distractors, solidifying its robustness and precision in real-world applications.

## 5.4 LIMITATIONS

While the DenseGrounding significantly improves the ego-centric 3D visual grounding task performance, it has limitations. In real-life applications, vague or ambiguous descriptions from human instructions pose challenges, as the model struggles without the necessary information to resolve ambiguities. The lack of clear instructions limits the effectiveness of the embodied agent. Future research should explore integrating human-agent interaction, allowing the model to query users for clarification, and improving adaptability and robustness in real-world scenarios.

## 6 CONCLUSION

In this work, we propose DenseGrounding, a novel approach to address the challenges of ambiguity in natural language descriptions and loss of fine-grained semantic features in multi-view 3D visual grounding. By leveraging LLMs for description enhancement and introducing the HSSE to enhance fine-grained visual semantics, our method significantly improves the accuracy and robustness of 3D visual grounding in ego-centric environments. Extensive experiments on the EmbodiedScan benchmark demonstrate that DenseGrounding outperforms existing state-of-the-art models, securing first place in the CVPR 2024 Autonomous Grand Challenge. These results highlight the importance of enriched language descriptions and effective cross-modal feature interactions for advancing embodied AI perception and enabling real-world applications in robotics and human-computer interaction.

## ACKNOWLEDGMENTS

The work is supported in part by the National Key R&D Program of China under Grant 2024YFB4708200 and the National Natural Science Foundation of China under Grant U24B20173.

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

# A   APPENDIX

CONTENTS

## A.1   MORE IMPLEMENTATION DETAILS

### A.1.1   ENHANCED BASELINE

First, we replace the RoBERTa (Liu et al., 2019) language encoder with the CLIP (Radford et al., 2021) encoder. Given that visual grounding requires a deep understanding of both linguistic instructions and environmental visual features, strong cross-modal alignment is critical for success. CLIP, specifically designed for such tasks, provides superior alignment between language and vision, making it a natural fit for this application. Moreover, visual grounding often involves identifying a diverse range of objects, and the original EmbodiedScan model relies on a detection pipeline to pretrain the visual feature encoders. However, this approach can be hampered by the long-tailed distribution of objects in the pretraining dataset, leading to suboptimal detection performance. To address this, we incorporate the Class-Balanced Grouping and Sampling (CBGS) method (Zhu et al., 2019) during pretraining, which has proven effective in mitigating data imbalance and enhancing detection accuracy across rare and common object categories.

### A.1.2   HYPERPARAMETERS

Our multi-view visual grounding model, DenseGrounding, is trained with the AdamW optimizer using a learning rate of 5e-4, weight decay of 5e-4, and a batch size of 48. The model is trained for 12 epochs, with the learning rate reduced by 0.1 at epochs 8 and 11. All other settings align with EmbodiedScan.

## A.2   PERFORMANCE ON INDIVIDUAL CLASSES

We show the performance of our method on the individual classes in the validation set in Fig. 4

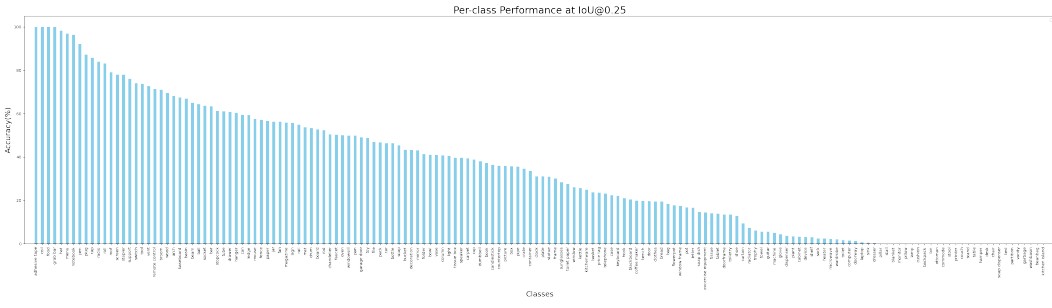

Figure 4: Performance of our method on each class in validation set of EmbodiedScan

## A.3 ANALYSIS ON LIMITED DATA SCENARIO

In Figure 5, we present a comparative analysis of the performance between our method and the baseline under varying amounts of training data. The results clearly demonstrate that our method consistently outperforms the baseline, especially in scenarios with limited data availability. Notably, when trained on just 40% of the training data, our method surpasses the baseline model that was trained on 80% of the data. This highlights the data efficiency and robustness of our approach, indicating its effectiveness even when training data is scarce.

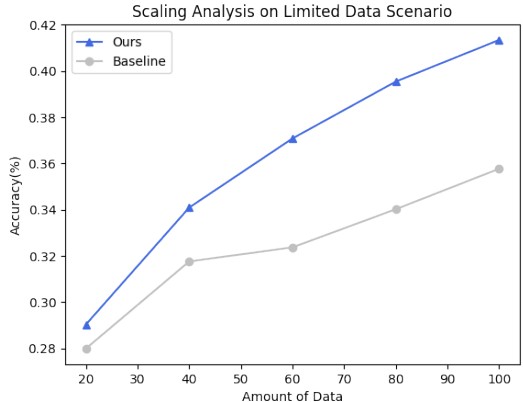

Figure 5: Comparison of DenseGrounding and EmbodiedScan on limited data scenario.

## A.4 ROBUSTNESS ANALYSIS

Table 6: Zero-shot Cross Dataset Performance

| Method | Training Scenes | Testing Scenes | Easy | Hard | Overall |
|---|---|---|---|---|---|
| EmbodiedScan | 3RScan+Matterport | Scannet | 9.95 | 7.94 | 9.81 |
| DenseGrounding | 3RScan+Matterport | Scannet | 18.57 | 12.42 | 18.14 |

Cross-dataset evaluation is essential for testing the generalization capabilities and robustness of egocentric 3D visual grounding methods. This process is particularly challenging due to differences in camera settings, scene layouts, and object characteristics in visual data across various datasets, which can significantly impact model performance. To further assess the zero-shot generalization of our approach, we reorganized the EmbodiedScan dataset to create a zero-shot setting. Specifically, we trained our model exclusively on scenes from 3RScan and Matterport3D and then evaluated its performance on ScanNet scenes, which were entirely unseen during training. This setup tests the model's ability to adapt to new environments with different camera settings and scene characteristics. As shown in Table 6, our method significantly outperforms the baseline in this zero-shot setting, demonstrating superior robustness and the ability to generalize effectively to new and diverse scenes despite the inherent cross-dataset challenges.

## A.5 IMPLEMENTATION DETAILS OF LSE

In this section, we provide the implementation details of the Language Semantic Enhancer (LSE) module, focusing on how the LLM is prompted. We delve into the specific prompts used and explain how information is selected to provide context to the LLM.

### A.5.1 PROMPT FOR LSE MODULE

In this section, we present the prompts used in the Language Semantic Enhancer (LSE) module, along with an example input and output, as illustrated in Fig. 6.

**[System]**

You are an AI spatial reasoning assistant specialized in rephrasing expressions related to 3D indoor scenes. Your task is to rephrase the given expression, enhancing the accuracy of spatial relationships based on the provided context.
Below are some expressions (visual grounding descriptions) that describe the scene and one specific expression that needs rephrasing. The text after LOCATION INFO provides the 3D position of the objects centers (unit: meters) in the lidar coordinate system.
Enhance this specific expression's accuracy and diversity by using more anchor points.

**NOTE:**
1. Input:

- Several visual grounding descriptions that provide reliable positional information. Each description includes the 3D coordinates of objects, the coordinates are given as (x, y, z) with units in meters.
- A visual grounding description that needs rephrasing.

2. Output: A rephrased visual grounding description.
3. Ensure that the target object remains unchanged.
4. Use the positional relationships of more objects in the scene to describe it, ensuring higher accuracy and without any potential ambiguity.
5. Only describe spatial relationships with high confidence; avoid adding extra information or assumptions.
6. Understand the spatial relationships between objects based on their 3D coordinates as a supplement to spatial information.
7. DO NOT provide the LOCATION INFO in the rephrased expression.
8. Just output the rephrased sentence directly, prohibit outputting any other statements.

**Example 1 (maybe not in this scene):**
Before rephrase:
select the pillow that is near the decoration. LOCATION INFO: pillow is located at (2.5, 2.8, 0.7). decoration is located at (1.25, 3.5, 0.5)

After rephrase:
Choose the pillow near the decoration, it is also situated beneath the mirror and positioned between the fireplace and the decoration

**Example 2 (maybe not in this scene):**
....

**[User]**
**Visual Grounding Descriptions:**

- choose the clock that is in front of the towel. LOCATION INFO: clock is located at (0.04, -1.85, 1.52). towel is located at (0.15, 1.97, 0.55)
- select the clock that is farthest from the pitcher. LOCATION INFO: clock is located at (0.07, -2.08, 1.85). pitcher is located at (-1.44, 1.50, 0.85)
- the clock that is closer to the chandelier. LOCATION INFO: clock is located at (0.04, -1.85, 1.52). chandelier is located at (0.24, 2.38, 2.20)
- ........

**Visual grounding description that needs rephrasing:**
the clock that is closer to the refrigerator. LOCATION INFO: clock is located at (0.04, -1.85, 1.52). refrigerator is located at (-1.05, 0.50, 0.76)
Now, rephrase it according to the above requirements.

**[Assistant]**
Choose the clock that is closer to the refrigerator, positioned beneath the chandelier and in front of the towel

Figure 6: Sample Input and Output of LLM for Language Enhancement

### A.5.2 Details on SIDB

**Context Description Selection.** Given a text description to be augmented (i.e., the raw description) and a SIDB containing text descriptions (i.e., context descriptions), we select k relevant context descriptions (we used k=50) to provide the LLM with scene and object relationship information. Specifically, when selecting descriptions as context to be utilized by the LLM for augmentation, we find descriptions in the SIDB that both belong to the same scene as the raw description and contain the target or anchor class of the raw description in their text. If more than 50 descriptions qualify, we randomly select 50 descriptions from among them. Conversely, if fewer than 50 descriptions qualify, we randomly select from the descriptions in the same scene. This strategy simplifies the process of context selection and allows for diversity in the context provided to the LLM.

