# OpenReview forum: "DenseGrounding: Improving Dense Language-Vision Semantics for Ego-centric 3D Visual Grounding"
_ICLR.cc/2025/Conference — ICLR 2025 Poster_

### Official Review · Reviewer_9Djk · 2024-11-02

**Soundness:** 3
**Presentation:** 3
**Contribution:** 2
**Rating:** 6
**Confidence:** 4

**Summary:**

This paper introduces a method for ego-centric 3D visual grounding, where agents are able to locate target objects in real-world 3D spaces based on verbal descriptions. The main goal of this paper is to address two critical challenges: 1. how to enhance the fine-grained visual semantics by improving the sparse fusion between ego-centric multi-view images, and 2. how to improve the textual semantic context in arbitrary language descriptions. To tackle these issues, the authors first introduce the Hierarchical Scene Semantic Enhancer, which effectively captures fine-grained scene-level global semantics. Secondly, they introduce the Language Semantic Enhancer, which uses LLM-assisted description augmentation to enhance natural language descriptions. The method presented in this paper achieves the SOTA result on the EmbodiedScan benchmark, demonstrating its effectiveness.

**Strengths:**

1. The motivation is clear, the improvement is substantial, and the paper is easy to follow.
2. They enhance the language description quality on EmbodiedScan. By releasing the improved annotations, they can benefit future projects.

**Weaknesses:**

1. Section 4.2, 'Prompt for LLM-based Enhancement,' lacks implementation details, such as prompts, and needs more examples to demonstrate the enhanced descriptions as well as the statics about the enhanced annotations including amount, average length.
2. The methods used in the hierarchical scene semantic enhancer lack novelty. Similar attention-based methods, such as BUTD-DETR [1], have been widely used in previous works. The author should emphasize the difference.
3. Table 1 lacks proper citations.

[1] https://arxiv.org/pdf/2112.08879

**Questions:**

1. The details of selecting the top-k most related descriptions used in LSE are not listed in the paper;I am still confused about how to rank these descriptions.

---

> ### Author Response · Authors · 2024-11-21
>
> We would like to thank the reviewer for your insightful comments and valuable suggestions, which will help us improve our paper.
>
> ### `Q1 and W1 - Implementation Details for Prompt`
> **Table 1: Statistics of the Augmented Prompt**
> | |Mini|	Full|
> |-|-|-|
> |Total Augmented Data|	48k	|247k|
> |Avg. Text Length Before Aug.	|12.00|	12.75|
> |Avg. Text Length After Aug.|	26.92	|26.13|
>
> Thank you for your valuable suggestion. We acknowledge that adding more details in Section 4.2 would improve the clarity of the paper. To address this, we have updated the manuscript to include the exact prompts used, examples of the enhanced descriptions generated by the LLM  **(see Appendix A.5.1)**, and statistics such as the total number of augmented annotations and their average length in Table 1.
>
> Regarding the selection of the k most related descriptions in the LSE for augmenting an input text description, we select context descriptions from the SIDB that mention the same objects as the input description. If fewer than k such relevant descriptions are available, we randomly select additional descriptions from the same scene to introduce diversity. Further details can be found in **Appendix A.5.2**.
>
> ### `W2 – Novelty of the HSSE`
>
> **Table. 2. Ablation Study on the proposed modules**
> | Method| Easy@0.25 | Hard@0.25 | Overall@0.25|
> |-|-|-|-|
> |Baseline | 35.77 | 30.51 | 35.77|
> |Baseline + HSSE|39.58|32.60|39.02|
> |Baseline + LSE|39.67|34.49|39.25|
> |Baseline + LSE+HSSE (Ours)	|**41.95**|**34.38**|	**41.34**|
>
> Thank you for the comment. While attention mechanisms are widely utilized, our HSSE is uniquely tailored for the egocentric visual grounding task. Due to the sparse nature of egocentric views, each individual input offers only a partial glimpse and cannot fully capture the entire scene. The HSSE module is designed to effectively enhance interactions of multiview visual features across individual views, leading to a more comprehensive scene representation. It also facilitates improved cross-modal interactions between textual data and the refined, fine-grained visual features representing the scene. By specifically applying these methods to the challenges presented by sparse egocentric views, we address particular issues inherent in this domain. Moreover, as indicated in Table 2, integrating the HSSE module yields an overall performance improvement of 2.09% when used on top of LSE, and also achieving 2.09% in hard and 3.25% in overall performance when compared with the baseline method.
>
>
> ### `W3 – Missing Citations in Table 1`
>
> Thank you for bringing this to our attention. We have revised table in our paper to include proper citations for all referenced methods, ensuring appropriate acknowledgment of prior work.
>
> Moreover, we will be releasing the improved annotations and code to benefit future research.
>
> We sincerely appreciate your constructive feedback and will incorporate these improvements into our revised manuscript.

---

### Official Review · Reviewer_3qfg · 2024-11-02

**Soundness:** 3
**Presentation:** 2
**Contribution:** 2
**Rating:** 6
**Confidence:** 4

**Summary:**

In this paper, they propose a model for the ego-centric 3Dvisual grounding task where  the model finds a 9DoF 3D bounding box from multiple RGB-D images and a language description for the specified object. Based on the EmbodiedScan, they introduced two modules: Language Semantic Enhancer (LSE) and Hierarchical Scene Semantic Enhancer (HSSE). In LSE, they use the predicted bbox and label and a Scene Information Database (SIDB) for augmenting the input text. HSSE uses view-level semantic aggregation. It is notable that SIDB depends on the annotated dataset (see L. 312) during the construction. They achieved the best results in the EmbodiedScan dataset, although the detailed analyses are limited, compared with those performed in the previous EmbodiedScan paper.

**Strengths:**

1. Proposing Language Semantic Enhancer (LSE) for the text input augmentation. LSE depends on Scene Information Database (SIDB), which uses the scene annotation.
2. Proposing Hierarchical Scene Semantic Enhancer (HSSE).
3. The best results in the EmbodiedScan dataset although I am not sure the experimental settings are consistent with the existing models.

**Weaknesses:**

1. It seems that SIDB depends on the annotated dataset as explained in the L.312. Isn’t this cheating in the experiments with the EmbodiedScan dataset because the model can reach rich symbolic textual information? Did you perform some ablation study for this?
2. The model improvement from EmbodiedScan seems incremental: the improvement of
3. Some experimental settings seem still unclear from the paper. For example, it is not clear whether authors use rendered or real images in EmbodiedScan.
4.  It is also expected that authors explain what are the inputs of the previous models in the main result table of Table 1. For example, ScanRefer depends on the 3D point cloud (pcd) and text. BUTD-DETR uses RGB-D as noted in Table 5 of the EmbodiedScan paper.
5. The experimental setting of ego-centric 3D visual grounding is not fully explained in this paper. For example, it is not clear how they obtain the camera pose, e.g., intrinsic and extrinsic matrices are obtained from the writings of Section 3.1. If models depend on them, it is recommended that authors update manuscripts for explaining the experimental settings.
6. The experiments are limited in the EmbodiedScan dataset. It is also interesting whether this method can be expanded for related datasets and tasks.
7. Limited related studies. It is recommended that authors include a wide branch of 3D visual groundings and question answering tasks.

**Questions:**

1. Is it possible to construct SIDB without using the dataset annotations? DId you perform ablation for this?
2. Can you clarify whether you use real or rendered images for the experiments? Also can you provide the ablation study of real vs rendered images as of Table 7 of  EmbodiedScan paper?
3. How do you treat augmentation errors in LSE?
4. Is it possible to report the detailed results, e.g., object category-wise result, as of EmbodiedScan?

---

> ### Author Response · Authors · 2024-11-21
>
> ### `Q1 and W1 - Dependence on Annotated Dataset`
> We would like to clarify that the SIDB is constructed **exclusively from the training data, which is also used by baseline methods** (revised in L320-L321). This ensures a **fair comparison**, as all methods have access to the same level of information. Moreover, we have conducted an ablation study to examine the impact of different context information in SIDB's construction on model performance. Specifically, as presented in Table 2 of our paper, we compare our proposed solutions LLM+DB(R) (using text descriptions only) and LLM+DB(R+L) (using text descriptions with 3D locations)  against the baseline methods. Although our model's performance may degrade without location information, both versions of our method still significantly outperform all baselines.
>
> ### `Q2 and W3 - Use of Real or Rendered Images`
> Thank you for highlighting the need for clarity on whether we use real or rendered images in our experiments. We have updated our manuscript to include this detail. In our experiments, we follow the main experimental setting of **EmbodiedScan for ego-centric visual grounding** and use real images provided by the dataset.
>
> For the ablation experiments, the EmbodiedScan dataset does not supply rendered images nor provide details on how to obtain them. Therefore, we are unable to replicate the ablation study comparing real versus rendered images as presented in the EmbodiedScan paper. Additionally, the ablation study in EmbodiedScan concerning rendered images focuses on 3D Object Detection, which differs from our task of Egocentric Visual Grounding.
>
> ### `Q3 - Augmentation Errors in LSE`
> Thank you for your question. Indeed, augmentation errors are possible because current LLM models inherently have hallucination issues, making some degree of noise inevitable. However, based on our experiments, the results we have obtained so far are quite satisfactory, showing observable improvements of 3.48%. Furthermore, with potential advancements in LLM technology and hallucination problems being addressed, performance may continue to improve.
>
> ### `Q4 - Object Category-wise Result`
> Thank you for your interest in a detailed, object category-wise analysis of our results. Given that the egocentric multiview grounding task of the EmbodiedScan dataset contains 163 object categories in the validation set, including a detailed table for each class, it is impractical due to space limitations in the main paper. However, we have provided a comprehensive graph illustrating the performance of our method across these classes in **Appendix A.2** of our paper.
>
> ### `W4 - Clarification on Visual Inputs of Previous Models`
> We appreciate your feedback regarding the need to clarify the inputs used by previous models in Table 1. We have updated the caption and the main text to specify that, consistent with the experimental settings of EmbodiedScan, we and the baseline methods use RGB-D inputs only. This ensures that comparisons are fair and that **all methods are evaluated under the same conditions**.
>
> ### `W5 - Clarification on Camera Pose and Experimental Settings`
> The EmbodiedScan dataset provides the camera's intrinsic and extrinsic parameters, which include camera poses and calibration data. Similar to the baseline methods in EmbodiedScan, we utilize this readily available information in our method. We have updated Section 3.1 of our manuscript to explicitly mention the use of these camera parameters and to explain the experimental settings in greater detail.
>
> ### `W6 - Applicability to Other Datasets and Tasks`
> Table 1. Performance of LSE on Scanrefer
> |Method|	Easy@0.25	|Hard@0.25	|Overall@0.25|
> |-|-|-|-|
> |EDA [1]	|74.02	|44.28|	52.06|
> |Ours (LSE only)|	75.34	|46.38	|54.01|
>
> We appreciate your interest in the potential expansion of our method to other datasets and tasks. The LSE is designed to be a versatile component that can be applied to similar tasks beyond the EmbodiedScan dataset. Its ability to augment training data through language models makes it suitable for other related tasks where textual and visual information interplay is crucial. As demonstrated in Table 1, **our proposed LSE module outperforms the baseline method EDA**, demonstrating the possible extension of our method to more tasks.
>
> ### `W7 - Inclusion of Additional Related Works`
> Thank you for your recommendation to include a broader range of related studies. We agree that providing a comprehensive overview of related work is essential for contextualizing our contributions. In response, we have updated the Related Work section of our paper to include additional relevant studies, encompassing recent advancements in 3D visual grounding, visual question answering, and related domains.
>
> **Reference**:
>
> [1] Wu et al. EDA: Explicit Text-Decoupling and Dense Alignment for 3D Visual Grounding. CVPR 2023.

---

> > ### Comment · Reviewer_3qfg · 2024-11-26
> > **OK**
> >
> > I have read the author response that resolve some concerns on this paper and I have updated my scoring.

---

> > > ### Author Response · Authors · 2024-11-26
> > >
> > > We are delighted that the revisions and updates to our paper have effectively addressed the concerns you raised. Your careful examination and constructive comments have been invaluable in helping us improve the clarity, precision, and overall quality of our paper.
> > >
> > > We deeply appreciate your thoughtful guidance and the recognition of the improvements made. We are grateful for the opportunity to refine our work based on your detailed suggestions.

---

### Official Review · Reviewer_RUeb · 2024-11-02

**Soundness:** 3
**Presentation:** 3
**Contribution:** 3
**Rating:** 6
**Confidence:** 4

**Summary:**

This paper proposes **DenseGrounding**, an interesting approach aimed at enhancing ego-centric 3D visual grounding with focus on two challenges

- Loss of Fine-Grained Visual Semantic: Sparse fusion of point clouds and multi-view images leads to semantic loss, which can hinder object detection in 3D space.
- Limited Texture Semantics: Ego-centric 3D visual grounding often has sparse or ambiguous language descriptions, which can lead to issues in object localization.

And this paper introduces two components:

- **Hierarchical Scene Semantic Enhancer (HSSE)**: This module enhances visual semantics by incorporating multi-view, scene-level information and facilitating cross-modal (text and visual) alignment.
- **Language Semantic Enhancer (LSE)**: Leveraging large language models (LLMs) with a structured Scene Information Database (SIDB), this component augments language descriptions with more contextually rich, diverse inputs.


They obtained strong results and the experiments are solid.

**Strengths:**

1. This paper's writing and motivations are both strong.

2. HSSE module for visual enhancement and LSE for text enhancement addresses two crucial limitations and make the proposed algorithm  effective towards real-world ego-centric tasks.

3. This proposed methods achieved competitive results on the EmboidiedScan benchmark.

**Weaknesses:**

- I am not too surprised to see DenseGrounding shows improved results on EmbodiedScan, a domain-specific dataset. However, the performance drop on the “hard” split in the HSSE ablation suggests limitations in handling complex scenes, raising doubts about whether it genuinely enhances spatial understanding or is just tuned to EmbodiedScan’s data characteristics. Testing on diverse datasets could clarify its robustness.
- Lack of Scaling Analysis: While the paper compares models trained on different data sizes, it lacks a detailed scaling law analysis. Understanding how performance scales with varying data sizes is crucial for practical applications, especially in data-scarce environments. A scaling study would show if DenseGrounding efficiently learns from limited data.
- Dependency on Complete Scene Graphs: The Language Semantic Enhancer (LSE) assumes access to a complete, accurate scene graph, which is rarely available in real-world settings. While using a predicted scene graph could mitigate this, the potential noise and errors in such predictions could degrade performance. Exploring the model’s robustness with imperfect scene graphs would enhance its practical relevance.

**Questions:**

- Could you evaluate the model on additional benchmarks to assess its generalization, and report on its zero-shot performance?
- Could you provide an analysis of the model’s performance across varying data sizes, compared to the baseline?

---

> ### Author Response · Authors · 2024-11-21
>
> Thank you for acknowledging the clear motivations and effectiveness of our HSSE and LSE modules in addressing key challenges and achieving competitive results on the EmbodiedScan benchmark. Regarding your concerns, we have provided our detailed responses below.
>
> ### `W1 and Q1 Generalization and Robustness: "Could you evaluate the model on additional benchmarks to assess its generalization, and report on its zero-shot performance?"`
> **Table 1: Cross-dataset zero-shot performance of Ego-centric 3D Visual Grounding Methods**
> |Method	|Easy@0.25	|Hard@0.25	|View-Dep@0.25|	View-Indep@0.25|	Overall@0.25|
> |-|-|-|-|-|-|
> |EmbodiedScan	|9.95	|7.94	|8.55|10.47|	9.81|
> |DenseGrounding|**18.57**|**12.42**|**17.16**|**18.65**|**18.14**|
>
> Thank you for the comment. We would like to clarify that the training and validation scenes in EmbodiedScan are collected from different scenes in three distinct 3D datasets (3RScan, Matterport3D, Scannet), and ensure that the scenes in the training set are different from those in the validation set.
>
> To further assess the generalization and robustness of egocentric 3D visual grounding methods, we have constructed a zero-shot setting within the EmbodiedScan dataset by reorganizing its samples. Specifically, we used the scenes from 3RScan and Matterport3D for training and then evaluated the model on the ScanNet scenes, which were not seen during training. The results, as shown in Table 1, indicate that our method **significantly outperforms the baseline method**, demonstrating the robustness of our approach in new and diverse scenes.
>
> ### `W2 and Q2 Scaling Analysis:`
> **Table 2: Limited Data Scenario Comparison on Overall@0.25**
> |Amount of Training Data|EmbodiedScan|DenseGrounding|
> |-|-|-|
> |20%|27.99|29.03|
> |40%|31.76|34.09|
> |60%|32.37|37.08|
> |80%|34.02|39.54|
> |100%|35.77|41.34|
>
>
> Thank you for highlighting the importance of understanding how performance scales with varying data sizes, especially in data-scarce environments. In response, we have conducted a scaling study by training our model on varying amounts of data by subdividing the mini-training set into smaller subsets. As illustrated in Table 2, our analysis shows that DenseGrounding efficiently learns from limited data and consistently outperforms the baseline across different data sizes. **Notably, when trained on just 40% of the training data, our method surpasses the baseline model that was trained on 80% of the data**. This highlights the data efficiency and robustness of our approach, indicating its effectiveness even when training data is scarce. We have included detailed results and discussions of this scaling study in the revised manuscript's appendix to provide clearer insights into the model's scalability.
>
> ### `W3 Dependency on Complete Scene Graphs`
> We would like to clarify that our method does not rely on an explicit scene graph from the dataset. Instead, LSE is constructed using text descriptions and 3D locations, **similar to the data used by other egocentric visual grounding baselines during training**. We provide these text descriptions and bounding boxes to an LLM, allowing it to learn object relations from the provided information to augment the training data without relying on predefined scene graphs. We will clarify this aspect in the revised paper to address any potential confusion.
>
> We sincerely thank you for your valuable feedback. We are committed to addressing these points to strengthen our work and provide a more comprehensive evaluation of our model, and we welcome any further discussions or clarifications.

---

> > ### Comment · Reviewer_RUeb · 2024-11-26
> >
> > I greatly appreciate the authors’ timely feedback, their extensive efforts in constructing this dataset, and their attempts to develop a better 3D grounding framework. As a result, I have decided to raise my score.

---

> > > ### Author Response · Authors · 2024-11-26
> > >
> > > Thank you for your thoughtful and constructive feedback, as well as for recognizing our efforts in developing a more robust 3D grounding framework. We have carefully incorporated your valuable suggestions to enhance the clarity and quality of our paper while providing additional information that we hope will benefit future research in this field.
> > >
> > > We deeply appreciate the time and effort you have dedicated to reviewing our work and are delighted to hear that our revisions have positively influenced your evaluation

---

### Official Review · Reviewer_1bvp · 2024-11-03

**Soundness:** 2
**Presentation:** 3
**Contribution:** 2
**Rating:** 6
**Confidence:** 5

**Summary:**

This paper aims to address the challenges of loss of fine-grained visual semantics and limited textual semantic context in ego-centric 3D visual grounding. It proposes a hierarchical scene semantic enhancer to enrich visual semantics, while leveraging a large language model (LLM) to enhance context and provide diverse textual semantics. Experimental results demonstrate that the proposed DenseGrounding achieves state-of-the-art results.

**Strengths:**

- The paper is well-organized and facilitates ease of comprehension.
- The paper introduces an improved baseline, and DenseGrounding achieves a significant performance enhancement.
- The idea of enhancing visual and linguistic features is reasonable; however, there is a potential risk of data leakage (which will be discussed later).

**Weaknesses:**

- Potential unfair comparisons and prior information leakage. From the ablation study in Table 3, it can be observed that the primary performance improvement of the method derives from the LSE module, which constructs a database and utilizes an LLM for enhancement. However,
    1. The construction of the database may leak crucial dataset priors, as it involves prior knowledge of other contextual objects in the scene and their relationships and localization information, even within the training set. This information should ideally be learned and acquired automatically through ego-centric multiview images.
    2. The database essentially constructs local contextual information between objects, which can significantly enhance the performance of 3D object detection. However, this information is extracted offline and then used as a shortcut to assist DenseGrounding.
    3. Minor: the use of the LLM introduces additional costs and lacks flexibility.
- The performance improvement of the HSSE module is modest, particularly in hard cases, where it may even have a negative impact.
- The novelty of the HSSE module is somewhat limited, as utilizing multi-scale features and employing vision-language fusion for enhancing visual features are common techniques in the vision-language domain.
- The description of the construction and selection of scene information in the dataset is not sufficiently specific. It is necessary to clarify what types of contextual information are being utilized, such as how each contextual object is defined or how specific objects are selected to consider their relationships.

**Questions:**

See weaknesses for details.

---

> ### Author Response · Authors · 2024-11-21
>
> We would like to express our gratitude for the detailed questions that could improve the quality of our paper.
>
> ### ` W1 "Potential unfair comparisons and prior information leakage..." `
> 1. Thank you for your comment. First of all, we understand your concern about potential data leakage and potential unfair comparison:
> - - **Preventing Potential Data Leakage**: In our work, we ensure that the **LLM does not have access to or utilize any information from the test set**. We would like to highlight that the database is constructed based solely on the training set, as you stated. During training, the LSE module uses LLM to augment the training only based on the database (SIDB) constructed from training data which is also used in training other baseline methods. During inference, the LSE module is not used to augment any test input data. This practice aligns with standard protocols to **prevent any unfair advantages** or leakage of test set priors. Additionally, we would like to emphasize that in EmbodiedScan the training and testing scenes are collected from entirely **separate** environments. Therefore, the scenes used during the evaluation were neither seen by the model during training nor incorporated into the LSE module.
> - - **Ensuring Fair Comparison**: Furthermore, **to ensure a fair comparison, we provided a baseline that also utilizes additional contextual objects from the training data.** It utilizes the additional context by concatenating multiple text samples describing the same target to form a new training set (**denoted as "Concat Sample" in Table 2 of the paper**). Under this experimental setting, our LSE module still **outperforms** this strong baseline.
>
> 2. **Concerns on SIDB and Cost**: Thank you for raising the point about the SIDB and the associated costs of using an LLM. We would like to clarify that the database (SIDB) is constructed offline solely to improve the training data as data augmentation. During inference, the LSE including SIDB is not used, so there is no additional cost or impact on flexibility at that stage. Regarding the cost of using the LLM, generating 1,000 samples incurs an approximate cost of $0.16, which we consider modest relative to the 3.48% improvement in overall accuracy achieved. This approach also reduces the need for extensive manual annotation, saving both time and labor. We will emphasize these points in the revised paper to address your concerns fully.
>
> ### ` W2 and W3 Performance and Novelty of HSSE `
>
> **Table. 1. Ablation Study on the proposed modules**
> | Method| Easy@0.25 | Hard@0.25 | Overall@0.25|
> |-|-|-|-|
> |Baseline | 35.77 | 30.51 | 35.77|
> |Baseline + HSSE|39.58|32.60|39.02|
> |Baseline + LSE|39.67|34.49|39.25|
> |Baseline + LSE+HSSE (Ours)	|**41.95**|**34.38**|	**41.34**|
>
> We would like to highlight that the HSSE module is specially designed for the ego-centric visual grounding task. Given the sparse egocentric views, the input data of each individual view has a partial view and cannot fully represent the entire scene. The HSSE module is designed to effectively enhance multiview visual feature interactions between each view to obtain a better representation of the scene, as well as further facilitate cross-modal interactions between text and obtained fine-grained visual features.
>
>  Our approach leverages these methods to address the unique challenges presented by sparse egocentric views. Furthermore, as demonstrated in Table 1 of this comment, the inclusion of the HSSE module yields a 2.09% improvement in performance on hard cases compared to the baseline method. When the HSSE module is applied in conjunction with the LSE module, the overall performance improves by 2.09%.
>
> ### ` W4 Details about SIDB is not sufficient `
> Thank you for pointing out the need for a more detailed description of the SIDB construction and the selection of relevant scene information. Specifically, the construction of SIDB involves collecting all the text descriptions (the text input of training samples, such as "Find the bin in front of the table") from each of the scenes in the training data, as well as the locations of the objects described in these descriptions. When augmenting a raw text description during training, we find 50 related text descriptions in the scene from the SIBD. We select these descriptions by finding the ones that describe the same instances as the raw text description. Then, we provide this information to the LLM and request an augmented output. We have included a more detailed explanation in Appendix A.5.
>
> We sincerely thank you for your valuable feedback and welcome any further clarifications or questions they may have.

---

> > ### Comment · Reviewer_1bvp · 2024-11-27
> > **Response to Rebuttal**
> >
> > Thank you for the response, and some concerns are addressed.
> >
> > However, my concerns regarding HSSE remain. From Table 1 in the response, it can be observed that "baseline+LSE" achieves a result of 34.49 on the hard split, but when HSSE is added ("baseline+LSE+HSSE"), the result drops to 34.38, a decrease of -0.11. This suggests that after incorporating LSE, HSSE has a negative effect on the hard splits.

---

> ### Author Response · Authors · 2024-11-29
>
> Thank you for your continued feedback and for highlighting concerns regarding the performance of our HSSE module on the hard split. We appreciate the opportunity to provide further clarification.
>
> The hard split specifically focuses on samples containing multiple instances (more than three) of the same object class, which requires strong textual understanding to disambiguate and accurately localize the target object. In our framework, the LSE module is designed to enhance textual context, making it particularly effective for these cases and explaining its strong performance on the hard split.
>
> In contrast, the HSSE module is aimed at improving fine-grained visual semantics from ego-centric views. Upon evaluating the metrics, we observed a performance difference of only 0.11% in the hard split, equating to 5 incorrectly classified samples out of all validating samples (more than 50k). Thus, this minor difference is considered comparable, especially when viewed alongside the overall improvement of 2.09% in Table 1. While HSSE may not directly improve performance on the hard split as significantly as LSE, its contribution to overall performance remains substantial.
>
>
> **Table 2. Ablation on Proposed Modules under IoU ≥ 0.5**
> | Method                          | Easy@0.5 | Hard@0.5 | Overall@0.5 |
> |---------------------------------|----------|----------|-------------|
> | Baseline + LSE                  |  16.00   |  13.88   |    15.83    |
> | Baseline + LSE + HSSE (Ours)    | **19.34**| **17.35**|  **19.18**  |
>
>
> We would like to sincerely thank you for your valuable feedback, which has prompted us to conduct a deeper exploration of this issue. In our **further analysis**, we hypothesize that the IoU threshold of 0.25 introduced by the EmbodiedScan benchmark may not necessarily require highly fine-grained predictions, which could explain why the performance of the HSSE module is not fully apparent under this metric. To better assess the impact of HSSE, we evaluate the model under a stricter IoU ≥ 0.5 threshold, where predictions and ground truth bounding boxes must overlap by more than 50%. This stricter threshold requires a more accurate prediction of bounding boxes, making it a more rigorous test of the model's ability to capture fine-grained visual details. Under this metric, HSSE demonstrates a significant improvement. As shown in Table 2, incorporating HSSE alongside LSE brings significant gains, with improvements of **3.47% on hard samples** and **3.35% in overall performance** under this stricter metric.
>
> These results underscore the strengths of the HSSE module in providing fine-grained representations of ego-centric views, highlighting its complementary effect when combined with the LSE module. By enhancing the model's ability to capture fine-grained visual cues, HSSE significantly improves the precision of the predicted bounding boxes. This aligns with our motivation for designing the HSSE module, which aims to refine the model's visual understanding.
>
> We would like to thank you once again for your insightful comments, which have helped us improve the clarity of our explanation and the overall presentation of our work.

---

> > ### Comment · Reviewer_1bvp · 2024-11-29
> > **Official Comment by Reviewer 1bvp**
> >
> > Thank you to the authors for further exploration and explanation. My concerns have been addressed, and I will increase my score to 6.

---

> > > ### Author Response · Authors · 2024-11-29
> > >
> > > Thank you for your thoughtful response. We greatly appreciate your feedback and are glad that our further exploration and clarification have addressed your concerns.

---

### Official Review · Reviewer_WKF8 · 2024-11-03

**Soundness:** 3
**Presentation:** 3
**Contribution:** 3
**Rating:** 6
**Confidence:** 3

**Summary:**

This paper introduces DenseGrounding, designed to address the issues of sparse fine-grained visual semantics and ambiguous language descriptions in ego-centric 3D visual grounding tasks. DenseGrounding includes two main modules: the Hierarchical Scene Semantic Enhancer (HSSE) and the Language Semantic Enhancer (LSE), each focused on enhancing semantics in the visual and language domains, respectively. The HSSE effectively captures fine-grained global semantics and promotes alignment between visual and textual inputs, while the LSE utilizes large language models (LLMs) to enrich language descriptions, increasing semantic richness and reducing inherent ambiguities. DenseGrounding achieves SOTA performance on the EmbodiedScan benchmark and secured first place in the CVPR 2024 Autonomous Driving Grand Challenge Track on Multi-View 3D visual grounding, underscoring its potential for real-world applications.

**Strengths:**

1. The proposed LSE module applies LLM to enrich text semantics, introducing a novel perspective to this task.
2. The proposed HSSE module effectively preserves fine-grained semantic features by first extracting semantic features from individual views and then broadcasting them across the entire scene, thus addressing the issue of semantic sparsity.
3. The paper provides a clear definition of the novel ego-centric 3D visual grounding task, making it accessible and easy to follow. The design of the method’s modules and the implementation of the ablation studies are also well-explained, allowing readers to easily understand the structure of DenseGrounding and the improvements achieved.
4. The proposed method achieves SOTA performance on the ego-centric 3D visual grounding task and earned first place in the CVPR 2024 Autonomous Driving Grand Challenge Track for multi-view 3D visual grounding, highlighting its potential for real-world applications.

**Weaknesses:**

1. The proposed method lacks descriptions for certain model details, such as the Multi-scale Fusion and Query Selection components in Figure 2.
2. The paper does not provide a sufficiently clear explanation of how the LLM leverages the SIDB to enrich the semantics of descriptions. For instance, what is the specific format of the location information in the SIDB, and how does the LLM utilize this location information?

**Questions:**

The proposed model performs excellently on the ego-centric 3D visual grounding task. However, I am curious whether it could also be applied to common 3D visual grounding tasks, such as the ScanRefer benchmark.

---

> ### Author Response · Authors · 2024-11-21
>
> We sincerely appreciate your recognition of the novelty and effectiveness of our proposed method for multi-view ego-centric visual grounding.
>
> ### `W1: "proposed method lacks descriptions for certain model details... `
>
> Thank you for your comment. Regarding your concerns, we have added more details about these adopted modules in our paper in Sec. 3.2.
>
> ### `W2: "The paper does not provide a sufficiently clear explanation of how the LLM leverages the SIDB to enrich the semantics of descriptions..."`
>
> Thank you for your insightful comment. To enable the LLM to effectively enhance a single input data, it is essential for the model to have richer contextual information about the scene. To achieve this, we constructed the Scene Information Database (SIDB), which compiles text-based information about each scene using input descriptions available in the training set. By leveraging the SIDB, the LLM gains access to detailed contextual information, allowing it to provide more informed augmentations. To better illustrate our method, we have included the prompt in Figure 6, offering a clearer and more detailed explanation. We hope this addition satisfactorily addresses your concern.
>
> ### `Q1: "...whether it could also be applied to common 3D visual grounding tasks..." `
>
> **Table 1. Performance of LSE on Scanrefer**
> | Method | Easy@0.25 | Hard@0.25 | Overall@0.25 |
> | - | -| - | -  |
> | EDA [1] | 74.02 | 44.28 | 52.06 |
> | Ours (LSE only) | **75.34** | **46.38** | **54.01** |
>
>
> Thank you for this suggestion. Regarding the two modules (HSSE and LSE) proposed by DenseGrounding, the HSSE module is specifically designed to address problems in ego-centric scenarios and does not apply to common 3D visual grounding tasks. However, the LSE module is applicable to common 3D visual grounding tasks. We used the LSE pipeline to augment the Scanrefer training set and presented the performance in Table 1. It can be observed that our method outperforms the previous SOTA in common 3D visual grounding.
>
> Thank you once again for your valuable feedback, which will help us further improve the clarity and quality of our work.
>
> **References**:
>
> [1] Wu et al. EDA: Explicit Text-Decoupling and Dense Alignment for 3D Visual Grounding. CVPR 2023.

---

> ### Comment · Reviewer_WKF8 · 2024-12-03
>
> Thanks for the detailed responses. My concerns have been addressed, and the implementation details of the paper have also been clarified. As a result, I will maintain my previous score of 6.

---

> > ### Author Response · Authors · 2024-12-03
> >
> > We sincerely thank you for your thoughtful feedback and for taking the time to review our responses. We are glad to hear that our clarifications and additional details in the revised manuscript addressed your concerns.

---

### Author Response · Authors · 2024-11-25

Dear Reviewers,

Thank you for taking the time to review our manuscript and provide valuable feedback. We deeply appreciate your insights and have carefully addressed all comments. We have revised the paper accordingly, as detailed below:

1. **Highlighting HSSE Differences**: We have emphasized the proposed HSSE method in comparison with previous attention-based approaches to highlight the key differences.
2. **Database Construction Details**: Additional details regarding the construction of the database using the training set have been included in the Appendix.
3. **Included LLM Prompt**: We have provided the prompt used by the language model along with an example of how it is used to augment training samples in the Appendix for your reference.
4. **Expanded Experimental Results**: We have provided more experimental results to demonstrate the effectiveness and generalizability of our proposed methods.
5. **Enhanced Preliminaries Section**: We have improved the preliminaries section by adding more information to provide a clearer understanding of the egocentric visual grounding setting and the model architecture.
6. **Added Related Works:** As suggested, we have incorporated more related work into 3D Visual Grounding (3DVG) and 3D Question Answering (3DQA) tasks.

Please let us know if you have any further questions or require additional clarification on any point. We are eager to engage in further discussion and would greatly appreciate your timely feedback on our revised manuscript as the deadline is approaching.

We look forward to your response.

---

### Author Response · Authors · 2024-12-04
**Rebuttal Summary and Acknowledgment of Reviewers' Feedback**

Dear Program Chairs, Area Chairs, and Reviewers,

We sincerely appreciate the reviewers' valuable suggestions and comments during the rebuttal period. The reviewers found our proposed work to be well-motivated, clearly presented, and highly effective in addressing the key challenges of egocentric 3D visual grounding tasks, demonstrating significant performance improvements.

1. **Novel Methodologies**: Reviewers WKF8 and RUeb commended the innovative approach of the Language Semantic Enhancer (LSE), which enriches text semantics, and the Hierarchical Scene Semantic Enhancer (HSSE), designed to address semantic sparsity in egocentric multiview 3D environments. These novel contributions were highlighted as both intriguing and impactful.
2. **State-of-the-Art Results**: The strong results achieved—state-of-the-art performance on both the EmbodiedScan benchmark and the 2024 Autonomous Driving Grand Challenge—were acknowledged by all reviewers (Reviewers 3qfg, WKF8, RUeb, 9Djk, 1bvp). These outcomes were noted as clear evidence of the method’s practical value and real-world applicability (Reviewers WKF8, RUeb).
3. **Clarity and Accessibility**: The paper’s clarity, structure, and methodological presentation were widely praised, making the content accessible and easy to follow (Reviewers WKF8, 1bvp, RUeb, 9Djk).

In response to the reviewers' comments, we have made several enhancements, including clarifying specific sections of the paper, enriching the Appendix with additional details, and incorporating new experimental results. These revisions are detailed in our responses to the reviewers.

We are delighted that the reviewers **unanimously recommended acceptance** of our work, assigning consistent scores **(6, 6, 6, 6, 6)**.

---

### Meta-Review · Area_Chair_AW3i · 2024-12-20

**Metareview:**

This paper presents an approach to improving the ego-centric 3D visual grounding (3DVG) task by enhancing both fine-grained visual semantics and textual semantic context. The authors achieve this through the design of two key components: the Hierarchical Scene Semantic Enhancer (HSSE) module and the Language Semantic Enhancer (LSE) module. Their method demonstrates state-of-the-art (SOTA) performance in ego-centric 3D visual grounding.

The initial reviewer ratings were 6, 5, 5, 5, and 6, with several concerns raised regarding model details, experimental setup, and specific aspects of the proposed method. Key issues included:

-	Insufficient details on the model architecture and experimental setup (WKF8, 3qfg, 9Djk)
-	Lack of clarity regarding the SIDB (WKF8, 1bvp, 3qfg)
-	The potential extension of the proposed method to scene-level 3DVG tasks, such as the ScanRefer dataset (WKF8, 3qfg)
-	Concerns about potential information leakage in the LSE module (1bvp, 3qfg)
-	Modesty in the performance and novelty of the HSSE module (1bvp, RUeb, 9Djk)
-	The analysis of scaling law (RUeb)
-	LSE's reliance on complete scene graphs (RUeb)
-	Requests for additional experiments, including extra benchmarks and varying data sizes (RUeb, 3qfg)
-	Missing citations of related work (3qfg)

The authors have addressed these concerns thoroughly in their rebuttal, providing clarifications and additional explanations that have led to a positive shift in reviewer ratings. As a result, reviewers 1bvp, RUeb, and 3qfg have increased their scores to 6.

Given the overall positive feedback, the novelty and effectiveness of the proposed method, and its superior performance on the ego-centric 3DVG task - a key challenge for embodied AI - the AC recommends acceptance of this paper. The authors are encouraged to include the additional experiments discussed in the rebuttal in the final version.

**Additional Comments On Reviewer Discussion:**

The initial reviewer ratings were 6, 5, 5, 5, and 6, with several concerns raised regarding model details, experimental setup, and specific aspects of the proposed method. Key issues included:

-	Insufficient details on the model architecture and experimental setup (WKF8, 3qfg, 9Djk)
-	Lack of clarity regarding the SIDB (WKF8, 1bvp, 3qfg)
-	The potential extension of the proposed method to scene-level 3DVG tasks, such as the ScanRefer dataset (WKF8, 3qfg)
-	Concerns about potential information leakage in the LSE module (1bvp, 3qfg)
-	Modesty in the performance and novelty of the HSSE module (1bvp, RUeb, 9Djk)
-	The analysis of scaling law (RUeb)
-	LSE's reliance on complete scene graphs (RUeb)
-	Requests for additional experiments, including extra benchmarks and varying data sizes (RUeb, 3qfg)
-	Missing citations of related work (3qfg)

The authors have addressed these concerns thoroughly in their rebuttal, providing clarifications and additional explanations that have led to a positive shift in reviewer ratings. As a result, reviewers 1bvp, RUeb, and 3qfg have increased their scores to 6.

Given that all major concerns have been effectively resolved, the AC recommends acceptance of this paper.

---

### Decision · Program_Chairs · 2025-01-22

Accept (Poster)